

# Self-centeredness and selflessness: happiness correlates and mediating psychological processes

Michael Dambrun

Université Clermont Auvergne (UCA), LAPSCO CNRS, Clermont-Ferrand, France

## ABSTRACT

The main objective of this research was to test central assumptions from the Self-centeredness/Selflessness Happiness Model. According to this model, while self-centered psychological functioning induces fluctuating happiness, authentic–durable happiness results from selflessness. Distinct mediating processes are supposed to account for these relationships: afflictive affects (e.g., anger, fear, jealousy, frustration) in the case of the former, and both emotional stability and feelings of harmony in the case of the latter. We tested these hypotheses in two studies based on heterogeneous samples of citizens ($n = 547$). Factor analyses revealed that self-centeredness (assessed through egocentrism and materialism) and selflessness (assessed through self-transcendence and connectedness to other) were two distinct psychological constructs. Second, while self-centeredness was positively and significantly related to fluctuating happiness, selflessness was positively and significantly related to authentic–durable happiness. Finally, distinct psychological processes mediated these relationships (study 2). On one hand, the relationship between self-centeredness and fluctuating happiness was fully mediated by afflictive affects. On the other hand, emotional stability and the feeling of being in harmony partially mediated the relation between selflessness and authentic–durable happiness.

## INTRODUCTION

With the rise of positive psychology, the study of subjective well-being and happiness has emerged as a research topic of primary importance in psychology (e.g. *Lyubomirsky, Sheldon & Schkade, 2005*). In this perspective, *Dambrun & Ricard (2011)* proposed the Self-centeredness/Selflessness Happiness Model (SSHM). The objective of the present study was to test three main assumptions of the SSHM: (1) self-centeredness and selflessness are not simple opposites, but are distinct psychological constructs; (2) sustainable authentic happiness would be promoted by a selfless style of functioning, while fluctuating happiness would be induced by self-centered functioning; and (3) independent processes would mediate these specific relationships: emotional stability and feeling of being in harmony in the case of the former, and afflictive affects in the case of the latter. Let us now see in more detail the theoretical underpinnings of these hypotheses.

Corresponding author
Michael Dambrun,
michael.dambrun@uca.fr

## Self-centeredness and selflessness

The SSHM posits that happiness is intimately linked to self-consciousness states (*Dambrun, 2016*). Based on the dichotomy between the minimal self and the narrative self (e.g., *Gallagher, 2000*), the SSHM specifies the psychological processes that arise from self-consciousness states and, ultimately, happiness (*Dambrun & Ricard, 2011*; *Dambrun, 2016*). According to *Dambrun & Ricard (2011)*, at least two qualities of happiness are connected and under the influence of two types of psychological functioning: the self-centered psychological functioning (Self-centeredness) and the selfless psychological functioning (Selflessness). There is little likelihood that these two types of functioning can operate simultaneously, the two having a tendency to be in conflict and in opposition to each other. However, they are proposed to consist of specific characteristics, which generate qualitatively different processes. For this reason, it is predicted that these two types of functioning would emerge on two distinct dimensions. The first goal of this study was to verify the bi-dimensionality of self-centeredness and selflessness. In statistical terms, the SSHM predicts a two factor solution in a factor analysis rather than a single factor solution, with characteristics assessing self-centeredness loading on one dimension and those assessing selflessness loading on a second factor (hypothesis 1).

### The self-centered psychological functioning and the mediating role of afflictive affects

The perception of the self as a real entity with sharp boundaries underlies a self-centered psychological functioning. *Laborit (1979)* proposes that each entity (or organized structure in the form of an entity), which aims at its preservation, is led to favor gratifications that positively reinforce it, and to avoid disagreeable things that threaten its homeostasis. A self-centered functioning, which is intimately related to egoism, egocentrism, an exaggerated importance given to the self and ego-inflation (through material possessions for example), favors and strengthen the "hedonic principle" (e.g., *Higgins, 1997*). According to this principle, individuals are motivated to obtain pleasure (i.e., approach) and to avoid displeasure (i.e., avoidance). Attaining these objectives (i.e., obtaining gratification and avoiding disagreeable stimuli) creates a feeling of transitory pleasure, joy and satisfaction. However, these stimulus-driven pleasures are contingent upon the appearance or disappearance of certain stimuli (*Wallace & Shapiro, 2006*). The experience of pleasure is, by nature, fleeting and dependent upon circumstances. It is unstable and the sensations it evokes soon become neutral (i.e., hedonic adaptation, e.g., *Brickman, Coates & Janoff-Bulman, 1978*; *Lyubomirsky, 2011*). In addition, the impossibility of attaining valued objectives gives rise to afflictive affects such as frustration, anger, hostility or jealousy, which damage well-being (e.g., *Miller et al., 1996*). Thus, by trying to maximize pleasures and avoiding displeasures, self-centeredness induces a fluctuating happiness in which phases of pleasure and displeasure alternate repeatedly (*Dambrun et al., 2012*). Momentary-level scores show that reported happiness varies significantly both with the time of day and day of week. While particular activities significantly increase happiness, others are associated with a decrease in happiness, with the result being that there are

important fluctuations in happiness over the course of a day or week (*Csikszentmihalyi & Hunter, 2003*). Under this perspective, *Dambrun et al. (2012)* developed a subjective fluctuating happiness scale (i.e., the SFHS), assessing perceived degrees of variation in happiness. According to the SSHM, self-centeredness would be positively and significantly related to this scale (hypothesis 2). Moreover, the relationship between self-centeredness and subjective fluctuating happiness would be mediated, at least partially, by afflictive affects (hypothesis 3).

Some results provide preliminary support for these predictions. First, *Dambrun et al. (2012)* found that self-centered values such as Schwartz's self-enhancement values (i.e., achievement and power, e.g., *Schwartz, 1992*; *Schwartz, 2003*) were related to subjective fluctuating happiness. In a second study, *Dambrun & Ricard (2012)* revealed that a scale assessing afflictive affects and composed of fifteen afflictive emotions (e.g., hostility toward other, jealousy, personal frustration) was positively and significantly related to the subjective fluctuating happiness scale. Thus, it is possible that afflictive affects mediate the relationship between self-centeredness and subjective fluctuating happiness.

### The selfless psychological functioning and the mediating role of both emotional stability and harmony feeling

The SSHM (*Dambrun & Ricard, 2011*) postulates the existence of a second self-based psychological functioning namely "selflessness". Unlike the self-centered functioning, "selflessness" is based on a weak distinction between self and others, and between self and the environment as a whole, which takes the form of a sense of connectedness (e.g., *Leary, Tipsord & Tate, 2008*). Selflessness is closely related to self-transcendence (e.g., *Cloninger, Svrakic & Przybeck, 1993*; *Levenson et al., 2005*; *Piedmont, 1999*). Selflessness would be primarily guided by the principle of "harmony", in the sense that the individual is facing a harmonious adjustment[1] with the different elements of the environment, including others and the different forms of life (*Leary, Tipsord & Tate, 2008*), as well as their deep personal aspirations. This principle would guide all psychological activities (i.e., conation, motivation, attention, emotion, cognition, behavior). For example, on an emotional level, the principle of harmony is linked to emotions of benevolence, as empathy, compassion or deep respect. Several studies show that pro-social behaviors, which are influenced by benevolent affects (e.g., *Dovidio & Penner, 2001*), are not only beneficial to others but also for the individual themselves. They are associated with better well-being (e.g., *Dambrun & Ricard, 2012*; *Weinstein & Ryan, 2010*), positive emotions (*Fredrickson et al., 2008*) and a reduction in psychological distress (*Carson et al., 2005*). Similarly, nature-connectedness is positively associated with well-being (e.g., *Howell et al., 2011*).

According to the SSHM, selflessness would be positively and significantly related to authentic–durable happiness. Authentic happiness is understood, here, as an optimal way of being; a state of durable contentment and plenitude or inner peace (based on a quality of consciousness, which underlies and imbues each experience, emotion and behavior, and allows us to embrace all the joys and the pain with which we are confronted). Under this perspective, Dambrun and his colleagues (*2012*) developed the Subjective Authentic–Durable Happiness Scale (SA-DHS) for assessing subjective authentic–durable happiness.

[1]Harmony means there is perfect agreement between the diverse parts of whole. When the self is perceived as being an interconnected element of the whole, a person's psychological functioning becomes more mindful and respectful of all the elements comprising this whole. Here the whole is meant in its broad sense. It is composed of the totality of elements, which make up our environment, including not only oneself and other human beings, but also all forms of natural life. In other words, the perception of interconnectedness underlies a specific psychological functioning by which the individual adjusts harmoniously to all of the elements (for more details, see *Dambrun & Ricard, 2011*).

As expected, they found that this scale was positively and significantly related to selfless values, such as benevolence and universalism (i.e., self-transcendence values, e.g., *Schwartz, 1992*; *Schwartz, 2003*). Interestingly for our purpose, using the Self-Transcendence Inventory (STI; *Levenson et al., 2005*) as a marker of selflessness, Dambrun and Ricard (*Dambrun & Ricard, 2012*) found that self-transcendence was positively and significantly related to subjective authentic–durable happiness. Thus, selflessness would be positively and significantly related to authentic–durable happiness. Because subjective authentic-durable happiness is theorized as a specific consequence of a selfless functioning, it is predicted they will be positively and significantly related even when self-centeredness will be statistically controlled for (hypothesis 4). Similarly, because subjective fluctuating happiness is theorized as a specific marker of a self-centered functioning, it is predicted that self-centeredness will be positively and significantly related to subjective fluctuating happiness even when selflessness will be statistically controlled for (hypothesis 2).

According to the SSHM, at least two processes would mediate the relationship between selflessness and authentic-durable happiness: (1) emotional stability and (2) the feeling of being in harmony.

First, benevolent affects, induced by the harmony principle, are not very sensitive to changes in the environment. As noted by *Sprecher & Fehr (2006)*: "compassionate love may be experienced for someone to whom love is not reciprocated" (p. 228). This type of affect would be relatively stable and would contribute to emotional stability and lasting happiness. Similarly, connectedness to others has been found to correlate modestly, but positively, with emotional stability (*Leary, Tipsord & Tate, 2008*). Several works have examined the relationship between emotional stability and well-being. In these studies, emotional stability is mostly assessed by scoring the neuroticism factor of the big five inventory in the opposite direction. For example, *Hills & Argyle (2001)* reported that emotional stability was the greater correlate of happiness among the big five factors. Similarly, in a meta-analysis, *DeNeve & Cooper (1998)* found that emotional stability was the strongest predictor of life satisfaction and happiness. On the basis of the SSHM, it is predicted that the relationship between selflessness and subjective authentic-durable happiness would be mediated, at least partially, by emotional stability (hypothesis 5).

In addition, *Dambrun & Ricard (2011)* propose that selflessness promotes the feeling of being in harmony with the environment (including others) and with oneself. This process also would mediate the relationship between selflessness and authentic-durable happiness. The perception of harmony, feelings of harmony and their relationships with happiness are present in various cultural traditions. Harmony has been approached as equilibrium, a state of homeostasis (*Headey & Wearing, 1989*). In the Yin-Yang theory, homeostasis is the ideal state for the entire universe, a state of harmony with the great natural principles (*Lu, 2001*). In psychological terms, it seems possible to distinguish three different levels of harmony: (1) being in harmony with oneself, (2) interpersonal or social harmony and (3) being in harmony with the world, including both the natural environment and the universe. These three levels are fundamental in numerous Asian philosophies (e.g., *Koller & Koller, 2007*). Harmony with oneself (e.g., Body–mind harmony or knowledge–feeling–idea harmony) has been proposed to be the essential characteristic of healthy and happy people

(see *Yan, 2006*), as well as of harmonious society (*Yaqin, 2008*). It is proposed that each level of perceived harmony (cognitive component; see *Kjell et al., 2015*) contributes to the feeling of being in harmony (affective component). Being in resonance with its profound aspirations can favor the feeling of being in harmony with oneself. Being connected to others and feeling empathy or compassion can promote the feeling of being in harmony with others. Being connected to the natural world and to the universe can lead to the feeling of being in harmony with the nature and the cosmos. These types of feeling are intimately linked with the characteristics of sustainable and authentic happiness as serenity or inner peace, a dimension of happiness that remains relatively unexplored (see *Dambrun et al., 2012*). Thus, on the basis of the SSHM, it is predicted that the feeling of being in harmony (affective component) would mediate, at least partially, the relationship between selflessness and subjective authentic–durable happiness (hypothesiss 6).

## Overview of the present research

In order to assess self-centeredness and selflessness, various measures were used. In the seminal theoretical paper of 2011 (*Dambrun & Ricard, 2011*), three main dimensions defined self-centeredness: a self-centrism bias (i.e., the self takes on a central point of reference with regard to various psychological activities), an exaggerated sense of importance given to the self (e.g., considering that one's own condition is more important than that of others and that this takes unquestionable priority) and an hedonic process (i.e., approach of gratifying stimuli and avoidance of disagreeable ones). Across two studies, a measure of egocentrism and a scale of materialism were used to assess self-centeredness. While the first measure has been previously developed to assess the first two dimensions of self-centeredness (i.e., self-centrality bias and an exaggerated sense of importance given to the self), the measure of materialism was used to assess the motivation to obtain material pleasures that is involved in the third dimension (i.e., hedonic process). Theoretically, selflessness has at least two important characteristics. First, its functioning is based on a weak distinction between self and others, and between self and the environment as a whole. Second, it is intimately related to self-transcendence; the process of seeing things as they are with clear awareness without strong distortion coming from biological and social conditioning (see *Dambrun & Ricard, 2011*). Therefore, a measure of self-transcendence and a scale assessing connectedness to other and to nature were incorporated in order to assess these two aspects of selflessness. While the connectedness scale allowed us to assess the first component of selflessness, the self-transcendence scale permitted us to measure the second. Because self-centeredness and selflessness tend to be in opposition, while also generating qualitatively distinct processes, a two factors solution with a moderate negative correlation between the two dimensions was predicted (hypothesiss 1). This prediction will be tested in two studies by means of factor analyses.

The second aim of the present research was to examine the relationships between self-centeredness/selflessness and subjective fluctuating happiness/subjective authentic–durable happiness (Hypotheses 2 and 4). Study 1 was mainly designed to examine these relationships.

**Table 1  Sample characteristics.**

| Characteristics | Study 1 | Study 2 |
|---|---|---|
| N | 243 | 304 |
| Population | Regional community | National community |
| Age mean in years (*SD*) | 40.5 (*17.3*) | 40.8 (*16.5*) |
| Age range in years | 18–87 | 17–86 |
| Female (%) | 53.0 | 62.2 |
| Religious believer (%) | 33.5 | 37.5 |
| Socio Economic Status (*SD*) | 2.9 (*.52*) | 2.9 (*.42*) |
| Education (*SD*) | 2.9 (*1.0*) | 2.6 (*2.1*) |

Notes.
SES was coded from 1 (extremely low SES) to 5 (extremely high SES). Similarly, education was coded from 1 (extremely low education) to 5 (extremely high education).

Finally, the third goal of the present research was to identify the underlying processes mediating these relationships (i.e., Hypotheses 3, 5 and 6). The mediation hypotheses will be specifically tested in study 2 by means of multiple mediation modeling (*Preacher & Hayes, 2008*).

# STUDY 1

This study was mainly designed to empirically test two assumptions: (a) Self-centeredness and selflessness are distinct psychological construct that are not simple opposites on a single continuum. This prediction was tested using a factor analysis. (b) Self-centeredness and selflessness would have specific happiness correlates. While subjective fluctuating happiness is expected to correlate positively and significantly with self-centeredness even when selflessness will be statistically controlled for, it is predicted that selflessness would be positively and significantly related to subjective authentic–durable happiness even when self-centeredness will be statistically controlled for.

## Method
### Participants and procedure
Two hundred and forty-three voluntary participants were recruited for the study on a voluntary basis through personal contact. The study took place in France. The sample was composed exclusively of adults from a regional community. They were adequately heterogeneous in age, gender, education, religiosity and socio-economic status (see Table 1). The questionnaire was delivered personally to each voluntary participant. All participants had 24 h to respond to the questionnaire and return it. This study consisted in an anonymous survey of consenting adults. Thus, the data were analyzed anonymously. This study has been approved by the Sud-EST VI statutory Ethics Committee (2014-CE36; IRB00008526), according to French legal requirements L. 1121-1-2 and R 1121-3.

### Materials
The questionnaire mainly comprised six scales: two scales assessing happiness (fluctuating happiness and authentic–durable happiness), two scales assessing selflessness (self-transcendence and connectedness to other and to nature), and two scales assessing

self-centeredness (egocentrism and materialism). At the end of the questionnaire, all the participants were asked to provide socio-demographic information (i.e., age, gender, education level, religiosity, and SES).

### Fluctuating happiness

The 10-item Subjective Fluctuating Happiness Scale (SFHS; see *Dambrun et al., 2012*) was used to assess fluctuating happiness. Participants had to indicate how much they agreed (7) or disagreed (1) with each of the 10 statements. A single composite score of subjective fluctuating happiness was computed by averaging responses to the 10 items (e.g., "My level of happiness is rather unstable, sometimes high, sometimes low", "I have times when I swing from moments of total bliss to much less satisfying moments"). The possible range of score is from 1.0 to 7.0, with higher scores indicating greater fluctuating happiness. The reliability of this scale was satisfactory ($\alpha = .92$).

### Authentic-durable happiness

The 16-item Subjective Authentic-Durable Happiness Scale (SA-DHS), developed by Dambrun and colleagues (*2012*), was used. Participants had to indicate their regular level of happiness in their life on a 7-point scale, ranged from 1 (very low) to 7 (very high). The scale comprised 16 items with 13 positively valenced items (e.g., happiness, bliss, overall well-being, serenity and plenitude), assessing authentic–durable happiness, and 3 negatively valenced items, only used to control for the compliance bias. A single composite score for subjective authentic–durable happiness was computed by averaging responses to the 13 positively valenced items. The possible range of scores was from 1.0 to 7.0, with higher scores reflecting greater authentic–durable happiness. The reliability of this scale was satisfactory ($\alpha = .94$).

### Self-transcendence

The Adult Self-Transcendence Inventory (ASTI; *Levenson et al., 2005*; *Le & Levenson, 2005*) was used to assess self-transcendence. This instrument was adapted from English to French.[2] This scale comprised 10 items ("My peace of mind is not so easily upset as it used to be", "I feel that my individual life is a part of a greater whole"). Participants were asked the extent to which they agree with each statement compared to five years ago, on a 7-point Likert scale ranging from 1 (strongly disagree) to 7 (strongly agree). The reliability of this scale was satisfactory ($\alpha = .76$).

### Connectedness

To measure the level of participants' connectedness, the Allo-Inclusive Identity scale developed by *Leary, Tipsord & Tate (2008)* was used. This instrument was adapted from English to French.[2] This scale comprised 16 items assessing how people perceived "connection" or "relatedness" to other people (first dimension) and to the nonhuman natural world (second dimension). These two dimensions reflect two distinct subscales, each comprising 8 statements: connectedness to other (e.g., "the connection between you and your best friend of the other sex") and connectedness to natural world (e.g., "the connection between you and the earth"). For each statement describing the connection

[2]First, the scale was translated into French and then back translated. Second, in order to verify the internal consistency, prior to its incorporation into the present study, the scale was field-tested with a sample of 75 participants. In this field test, the reliability of the Adult Self-Transcendence Inventory (ASTI) was adequate ($\alpha = .89$), the reliability of each subscale of the Allo-Inclusive Identity scale also was satisfactory (connectedness to other, $\alpha = .77$; connectedness to natural world, $\alpha = .86$), and the 9 items from the material value scale also provided an adequate internal reliability ($\alpha = .77$).

between the respondent and some other entity, participants selected from a set of seven pairs of circles, one circle representing the respondent and the second circle representing the other entity (1 = widest separation between the two; 7 = closest overlap between the two). The reliability of each subscale was satisfactory (connectedness to other, $\alpha = .70$; connectedness to natural world, $\alpha = .85$).

### Egocentrism

To assess egocentrism, the scale developed by *Dambrun (2011)* was used. This scale comprised 10 statements (e.g., "The motto "me first, then the others" corresponds quite well to the way in which I behave", "I'm not someone who tries systematically to make the best personal profit of a situation (reverse coded)"). Participants were asked the extent to which they agree with each statement on a 7-point scale ranging from 1 (strongly disagree) to 7 (strongly agree). The reliability of this scale was satisfactory ($\alpha = .80$).

### Materialism

To assess materialism, nine items from the material value scale (MVS) proposed by *Richins (2004)* (see also *Richins & Dawson, 1992*) were used. This scale assesses three dimensions: possession-defined success (e.g., "Some of the most important achievements in life include acquiring material possessions"), acquisition centrality (e.g., "I usually buy only the things I need") and acquisition as the pursuit of happiness (e.g., "My life would be better if I owned certain things I don't have"). This scale was adapted from English to French[2]. Participants were asked the extent to which they agree with each statement on a 7-point scale ranging from 1 (strongly disagree) to 7 (strongly agree). The reliability of this scale was satisfactory ($\alpha = .76$).

All statistical analysis was performed using SPSS v.22.0 (IBM Corp., Armonk, NY, USA).

## Results

### Relationships between various measures

First, the means, standard deviations, and inter-correlations among all the measured variables were examined (see Table 2). First, while egocentrism and materialism correlated positively and significantly with fluctuating happiness (respectively $r = .40$ and $r = .21$), self-transcendence and connectedness to others were positively and significantly related to authentic–durable happiness (respectively $r = .34$ and $r = .23$) but not to fluctuating happiness. Materialism was not related to authentic–durable happiness. However, egocentrism and authentic–durable happiness were negatively correlated. In this study, connectedness to nature was not significantly related to happiness. Replicating previous studies, subjective fluctuating happiness and subjective authentic–durable happiness were significantly and negatively correlated ($r = -.50$). Finally and as predicted, while the main psychological constructs related to selflessness were positively and significantly related to each other (i.e., self-transcendence and connectedness to other), those assessing self-centeredness were also positively and significantly related to each other (i.e., egocentrism and materialism). Unexpectedly, connectedness to nature was not related to self-transcendence. For this reason, this measure was not included in further analyses.

**Table 2  Relationships between various variables (study 1; $n = 243$).**

|  | M | SD | 1 | 2 | 3 | 4 | 5 | 6 |
|---|---|---|---|---|---|---|---|---|
| 1. Self-Transcendence | 4.42 | .89 | – | | | | | |
| 2. Connectedness (Other) | 3.87 | .79 | .31*** | – | | | | |
| 3. Connectedness (Nature) | 2.64 | 1.15 | .31*** | .54*** | – | | | |
| 4. Egocentrism | 3.21 | .94 | −.29*** | −.16* | −.08 | – | | |
| 5. Materialism | 3.56 | .99 | −.16* | −.07 | −.04 | .38*** | – | |
| 6. Subjective Fluctuating Happiness (SFHS) | 3.95 | 1.30 | −.07 | −.04 | .07 | .40*** | .21*** | – |
| 7. Subjective Authentic-Durable Happiness (SA-DHS) | 4.24 | 1.02 | .34*** | .23*** | .05 | −.20** | −.09 | −.50*** |

Notes.
*** $p < .001$.
** $p < .01$.
* $p < .05$.

**Table 3  Relationships between self-centeredness/selflessness and happiness (study 1; $n = 243$).**

|  | Self-centeredness | | Selflessness | |
|---|---|---|---|---|
|  | β | Partial β (controlling for selflessness) | β | Partial β (controlling for self-centeredness) |
| Subjective fluctuating happiness (SFHS) | .36*** | .37*** | −.07 | .02 |
| Subjective authentic-durable happiness (SA-DHS) | −.17** | −.08 | .36*** | .33*** |

Notes.
*** $p < .001$.
** $p < .01$.

### Are self-centeredness and selflessness distinct constructs?

In order to test the predicted model, a factor analysis was performed. This analysis hypothesized that self-transcendence and connectedness to other formed one component assessing selflessness, and that this was distinct from a second factor assessing self-centeredness and composed of egocentrism and materialism. The principal component factor analysis with direct Oblimin rotation of the four scales disclosed two factors with Eigenvalues greater than 1. The first factor accounted for 42.5% of the explained variance and regrouped the measures of materialism and egocentrism (Eigenvalue = 1.7; factor loadings respectively .88 and .76). The second factor accounted for 25.9% of the explained variance and regrouped the measures of connectedness to others and the self-transcendence inventory (Eigenvalue = 1.04; factor loadings respectively .87 and .72).

### Are self-centeredness and selflessness related respectively to subjective fluctuating happiness and subjective authentic–durable happiness?

In order to examine the hypotheses regarding the relationships between self-centeredness, selflessness, subjective fluctuating happiness and subjective authentic–durable happiness, first the mean of self-centeredness was computed by averaging the measures of materialism and egocentrism. The mean of selflessness was computed by averaging the measures of connectedness to other and self-transcendence.[3] Then, standardized coefficients and partial regression coefficients were calculated using multiple regression analyses (see Table 3).

[3] The same basic findings were found using factor scores.

It was predicted that self-centeredness would be positively and significantly related to fluctuating happiness even when selflessness will be statistically controlled for. On the other hand, it was hypothesized that selflessness would be positively and significantly related to authentic–durable happiness even when self-centeredness will be statistically controlled for. Selflessness was positively and significantly related to authentic–durable happiness ($\beta = .36$, $p < .001$), but not to fluctuating happiness ($\beta = -.07$, $p > .28$). When self-centeredness was statistically controlled for, the relationship between selflessness and authentic-durable happiness still remained significant ($\beta = .37$, $p < .001$). Self-centeredness was significantly related to both fluctuating happiness ($\beta = .36$, $p < .001$) and authentic–durable happiness ($\beta = -.17$, $p < .01$). When selflessness was statistically controlled for, the relationship between self-centeredness and authentic–durable happiness vanished ($\beta = -.08$, $p > .18$), and the relationship between self-centeredness and fluctuating happiness still remained significant ($\beta = .33$, $p < .001$). Self-centeredness and selflessness were significantly and negatively correlated ($\beta = -.25$, $p < .001$).

## Discussion

As predicted on the basis of the SSHM, the various psychological constructs assessing selflessness and self-centeredness load on two distinct factors. While self-transcendence and connectedness to other load on the selflessness dimension, both egocentrism and materialism, appropriately, load on the self-centeredness factor. These two dimensions correlated negatively and modestly. They share approximately 6% of variance. These results confirm the first hypothesis and are compatible with the SSHM position that selflessness and self-centeredness are not simple opposites, but rather are distinct modes of psychological functioning that tend to be opposite (i.e., partial opposite).

If selflessness and self-centeredness are distinct modes of psychological functioning, they would result in distinct outcomes. This is exactly what it is found when the relationships between these modes of psychological functioning and happiness were examined. More precisely, selflessness was positively and moderately related to subjective authentic–durable happiness even when self-centeredness was statistically controlled for. Selflessness was not related to subjective fluctuating happiness. On the other hand, when selflessness was statistically controlled for, self-centeredness was positively and moderately related to subjective fluctuating happiness, but not to subjective authentic-durable happiness. Thus, both selflessness and self-centeredness seem to have their unique happiness marker, subjective authentic–durable happiness in the case of the former and subjective fluctuating happiness in the case of the latter. These results confirm the hypotheses 2 and 4.

## STUDY 2

This second study was mainly designed to identify the mediating processes by which selflessness is positively related to subjective authentic–durable happiness and by which self-centeredness is positively related to subjective fluctuating happiness. To the extent that reproducibility has been identified as a strong issue in psychology (see for example *Open Science Collaboration, 2015*), a second objective of study 2 was to test the reproducibility of the results obtained in study 1 by reproducing similar statistical analyses.

## Method

### Participants and procedure

Three hundred and four voluntary participants were recruited for the study on a voluntary basis through personal contact. The study took place in France. The sample was composed exclusively of adults from a national community. They were adequately heterogeneous in age, gender, education, religiosity and socio-economic status (see Table 1). The questionnaire was delivered personally to each voluntary participant via a paper booklet ($n = 153$) or an online survey platform ($n = 151$). All participants had 24 h to respond to the questionnaire and return it. This study consisted in an anonymous survey of consenting adults. Thus, the data were analyzed anonymously. This study has been approved by the Sud-EST VI statutory Ethics Committee (2014-CE36; IRB00008526), according to French legal requirement L. 1121-1-2 and R 1121-3.

### Materials

In addition to the measures used in study 1 (i.e., fluctuating happiness, authentic-durable happiness, self-transcendence, connectedness to other and to nature, egocentrism and materialism), three new scales were added. These new instruments assess the predicted mediating variables (i.e., afflictive affects, feeling of being in harmony, and emotional stability). At the end of the questionnaire, all the participants were asked to provide socio-demographic information (i.e., age, gender, education level, religiosity, and SES).

### Happiness

The 10-item Subjective Fluctuating Happiness Scale (SFHS; see *Dambrun et al., 2012*) was also used in this study. The reliability of this scale was satisfactory ($\alpha = .92$). The 16-item Subjective Authentic-Durable Happiness Scale (SA-DHS), developed by Dambrun and colleagues *(2012)*, was also used. The reliability of this scale was satisfactory ($\alpha = .95$).

### Self-Centeredness

As in study 1, both the material value scale (MVS), proposed by *Richins (2004)*, and the 10-item egocentrism scale, developed by *Dambrun (2011)*, were used. Their reliability was adequate (respectively $\alpha = .74$ and $\alpha = .78$).

### Selflessness

Again, the 10-item Adult Self-Transcendence Inventory (ASTI, *Levenson et al., 2005*; *Le & Levenson, 2005*) and the Allo-Inclusive Identity scale, developed by *Leary, Tipsord & Tate (2008)*, were incorporated in the questionnaire. The reliability of the ASTI was satisfactory ($\alpha = .79$). The reliability of each subscale of the Allo-Inclusive Identity scale also was adequate (connectedness to others, $\alpha = .79$; connectedness to natural world, $\alpha = .90$).

### Mediating variables

Three scales assessing the predicted mediating processes were added to the questionnaire: afflictive affects, feeling of being in harmony and emotional stability. Each scale was presented separately.

### Afflictive affects

To assess afflictive affects, the scale developed by *Dambrun & Ricard (2012)* was used. This scale comprised 15 statements (e.g., "hostility toward other" jealousy", "personal frustration", "angry", "fear", "threat"). Participants had to indicate their regular level for each type of affect on a 7-point scale ranged from 1 (very low) to 7 (very high). The reliability of each subscale was satisfactory ($\alpha = .86$).

### Feeling of being in harmony

To assess the feeling of being in harmony, a recent scale developed by Dambrun (2016, unpublished data) was used. This scale consists in 25 statements assessing harmony feeling in various domains (e.g., "I feel a perfect harmony between my ideals and my current life", "when I observe the sky during the night, I often feel a feeling of harmony", "I feel that my personal relationships with the person that share my life are almost harmonious", "I feel that my interpersonal relationships are rarely harmonious (reverse coded)"). Participants were asked the extent to which they agree with each statement on a 7-point scale ranging from 1 (strongly disagree) to 7 (strongly agree). The reliability of this scale was satisfactory ($\alpha = .86$). Since, in this study, assessing the feeling of being in harmony was the main interest, the Harmony in Life Scale of Kjell and colleagues (*2015*), which is more focused on the cognitive component of harmony (e.g., "I am in harmony"), was not used.

### Emotional stability

To assess emotional stability, the neuroticism factor of the big five inventory was measured and scored in the opposite direction, such that higher scores indicate greater emotional stability (for a similar methodology see, for example, *Hills & Argyle, 2001*). Specifically, the French version of the big five inventory, developed and validated by *Plaisant et al. (2010)*, was used. Participants had to indicate the extent to which they approve each statement on a 5-point scale ranging from 1 (strongly approve) to 5 (strongly disapprove). The reliability of this scale was adequate ($\alpha = .84$).

All statistical analysis was performed using SPSS v.22.0 (IBM Corp., Armonk, NY, USA).

## Results

### Relationships between various measures

First, the means, standard deviations, and inter-correlations among all the measured variables were examined (see Table 4). While egocentrism and materialism correlated positively and significantly with fluctuating happiness (respectively $r = .17$ and $r = .17$), self-transcendence and connectedness to other were positively and significantly related to authentic–durable happiness (respectively $r = .45$ and $r = .34$), but not to fluctuating happiness. As expected, subjective fluctuating happiness and subjective authentic–durable happiness were significantly and negatively correlated ($r = -.27$).

It was examined how the predicted mediating variables correlated with other measures. As predicted, afflictive affects mainly correlated positively and significantly with egocentrism ($r = .44$), materialism ($r = .20$) and subjective fluctuating happiness ($r = .36$). As expected, feelings of being in harmony were positively and significantly related to self-transcendence ($r = .39$), connectedness to others ($r = .42$) and to nature ($r = .47$),

Dambrun (2017), *PeerJ*, DOI 10.7717/peerj.3306

**Table 4  Relationships between various variables (study 2; $n = 304$).**

| | M | SD | 1 | 2 | 3 | 4 | 5 | 6 | 7 | 8 | 9 |
|---|---|---|---|---|---|---|---|---|---|---|---|
| 1. Self-Transcendence | 4.49 | .99 | – | | | | | | | | |
| 2. Connectedness (Other) | 4.10 | .93 | .37*** | – | | | | | | | |
| 3. Connectedness (Nature) | 3.05 | 1.36 | .25*** | .55*** | – | | | | | | |
| 4. Egocentrism | 2.93 | .95 | −.03 | −.05 | −.03 | – | | | | | |
| 5. Materialism | 3.39 | 1.00 | −.14* | .03 | −.05 | .26*** | – | | | | |
| 6. Subjective Fluctuating Happiness (SFHS) | 3.68 | 1.30 | −.02 | .00 | .04 | .17** | .17** | – | | | |
| 7. Subjective Authentic–Durable Happiness (SA-DHS) | 4.26 | 1.15 | .45*** | .34*** | .09 | −.09 | −.02 | −.27*** | – | | |
| *Mediating variables* | | | | | | | | | | | |
| 8. Afflictive affects | 3.04 | .92 | −.02 | .04 | .05 | .44*** | .20*** | .36*** | −.05 | – | |
| 9. Feeling of being in harmony | 4.31 | .78 | .39*** | .42*** | .47*** | −.06 | −.18** | −.23*** | .59*** | .00 | – |
| 10. Emotional stability | 3.07 | .86 | .34*** | .21*** | .09 | −.08 | −.12* | −.59*** | .55*** | −.29*** | .41*** |

**Notes.**
*** $p < .001$.
** $p < .01$.
* $p < .05$.
+ $p < .10$.
and subjective authentic–durable happiness ($r = .59$). However, significant negative correlations emerged between the measure of harmony feeling and both materialism ($r = -.18$) and subjective fluctuating happiness ($r = -.23$). The third mediating variable (i.e., emotional stability) was mainly related to self-transcendence ($r = .34$), connectedness to other ($r = .21$), subjective fluctuating happiness ($r = -.59$), subjective authentic–durable happiness ($r = .55$), afflictive affects ($r = -.29$) and feelings of being in harmony ($r = .41$).

Finally, while the main psychological constructs related to selflessness were positively and significantly related to each other (i.e., self-transcendence and connectedness to other), those assessing self-centeredness also were positively and significantly related to each other (i.e., egocentrism and materialism). Again, connectedness to nature was not related to self-transcendence. For this reason, this measure was not included in further analyses.

### Are self-centeredness and selflessness distinct constructs?

In order to further test the predicted model, again, a factor analysis was performed. This analysis hypothesized that self-transcendence and connectedness to others formed one component assessing selflessness, and that this was distinct from a second factor assessing self-centeredness and composed of egocentrism and materialism. The principal component factor analysis with direct Oblimin rotation of the four scales disclosed two factors with Eigenvalues greater than 1. The first factor accounted for 35.5% of the explained variance and regrouped the measures of connectedness to other and the self-transcendence inventory (Eigenvalue = 1.4; factor loadings respectively .84 and .82). The second factor accounted for 30.3% of the explained variance and regrouped the measures of materialism and egocentrism (Eigenvalue = 1.2; factor loadings respectively .80 and .77).

### Are self-centeredness and selflessness related respectively to subjective fluctuating happiness and subjective authentic-durable happiness?

Using the same procedure than in study 1, the relationships between self-centeredness (i.e., average of materialism and egocentrism), selflessness (i.e., average of connectedness to other and self-transcendence), subjective fluctuating happiness and subjective authentic–durable happiness were examined (see Table 5)[3]. As expected, selflessness was positively and significantly related to authentic–durable happiness ($\beta = .48$, $p < .001$), even when self-centeredness was statistically controlled for ($\beta = .48$, $p < .001$). Selflessness and fluctuating happiness were not significantly related ($\beta = -.02$, $p > .77$). Self-centeredness was positively and significantly related to fluctuating happiness ($\beta = .22$, $p < .001$), even when selflessness was statistically controlled for ($\beta = .22$, $p < .001$). Self-centeredness was not related to authentic–durable happiness ($\beta = -.07$, $p > .22$). Self-centeredness and selflessness were not significantly correlated ($\beta = -.07$, $p > .21$).

### Mediation analyses

In order to test the predicted multiple mediator models, the procedure advocated by *Preacher & Hayes (2008)* was followed (5,000 samples and confidence intervals = 95%). The averaged measure of self-centeredness (i.e., materialism and egocentrism) and selflessness (i.e., connectedness to other and self-transcendence)[3] were used. As previously mentioned, the measure of afflictive affects was significantly related to fluctuating happiness, but not to
**Table 5** Relationships between self-centeredness/selflessness and happiness (study 2; $n = 304$).

| | Self-centeredness | | Selflessness | |
| --- | --- | --- | --- | --- |
| | $\beta$ | Partial $\beta$ (controlling for selflessness) | $\beta$ | Partial $\beta$ (controlling for self-centeredness) |
| Subjective fluctuating happiness (SFHS) | 22*** | .22*** | −.02 | .00 |
| Subjective authentic-durable happiness (SA-DHS) | −.07 | −.03 | .48*** | .48*** |

Notes.
*** $p < .001$.

authentic–durable happiness. Feeling of harmony and emotional stability were significantly related to authentic–durable happiness, but also to fluctuating happiness (see Table 4). The scale of afflictive affects was significantly related to self-centeredness ($\beta = .40, p < .001$), but not to selflessness ($\beta = .01$). Emotional stability and feeling of harmony were positively and significantly related to selflessness (respectively, $\beta = .33, p < .001$ and $\beta = .49, p < .001$), but they were also significantly and negatively related to self-centeredness (respectively, $\beta = -.13, p < .02$ and $\beta = -.15, p < .01$).

Thus, the first predicted mediation model in which afflictive affects would mediate the relationship between self-centeredness and fluctuating happiness was tested. However, since both harmony feeling and emotional stability correlated significantly and negatively with both self-centeredness and fluctuating happiness, they were added as potential mediators in the *Preacher & Hayes (2008)* multiple mediation procedure with self-centeredness as an independent variable and fluctuating happiness as a dependent variable (see Fig. 1A). When afflictive affects, feeling of harmony and emotional stability were statistically controlled for, the relationship between self-centeredness and fluctuating happiness became non-significant ($b = .13, se = .08, p < .11$) and was significantly reduced ($b = .23, se = .07; z = 3.42, p < .001; CI: .09; .38$). As predicted, only afflictive affects significantly mediated this relationship ($b = .11, se = .04; z = 3.04, p < .003; CI: .04; .20$). Based on their bootstrap confidence intervals, harmony feeling and emotional stability did not mediate this relationship ($b = .00, se = .01; CI : -.03; .03$ for harmony feeling, and $b = .11, se = .03; CI : .00; .23$ for emotional stability). This model explains 39% of variance in term of subjective fluctuating happiness. Providing additional support, the results did not provide a stronger support for the reversed model, in which afflictive affects mediate the relationship between fluctuating happiness (IV) and self-centeredness (DV; $b = .08, se = .02; CI: .04; .12$), than for the predicted model ($b = .21, se = .05; CI: .12; .32$; see Fig. 1B).

The second predicted mediation model in which both feelings of harmony and emotional stability would mediate the relationship between selflessness and subjective authentic–durable happiness also was tested. Feelings of harmony and emotional stability were entered as mediators in a *Preacher & Hayes (2008)* multiple mediation procedure, with selflessness as an independent variable and authentic-durable happiness as a dependent variable (see Fig. 2). When feelings of harmony and emotional stability were statistically controlled for, the relationship between selflessness and authentic–durable happiness remained statistically significant ($b = .28, se = .07, p < .001$), but was significantly reduced

A

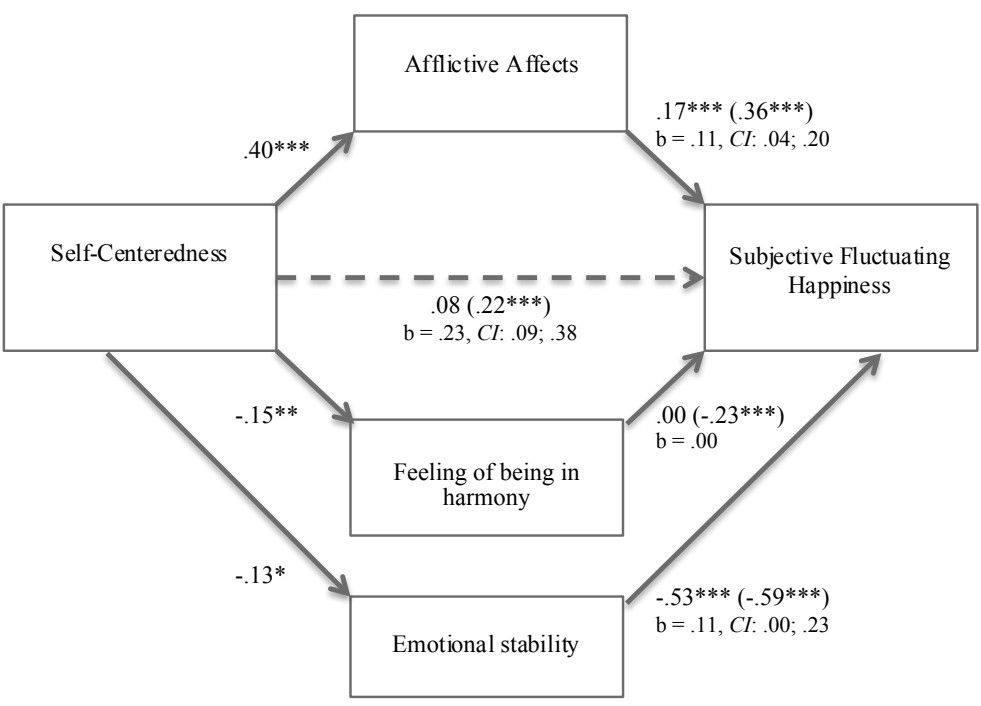

B

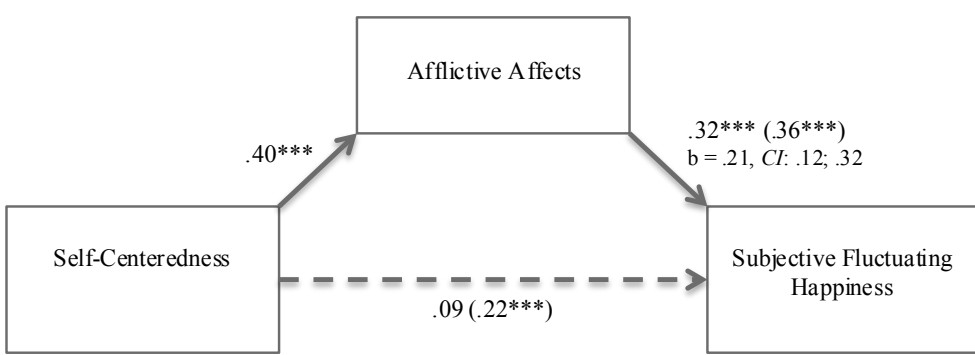

**Figure 1** **(A) and (B) Mediation model: afflictive affects as a mediator of the relationship between self-centeredness and subjective fluctuating happiness (study 2).**

($b = .41$, $se = .05$; $z = 7.55$, $p < .001$; CI: .31; .53). As predicted, both feelings of harmony ($b = .25$, $se = .04$; $z = 5.84$, $p < .001$; $CI$: .18; .35) and emotional stability ($b = .16$, $se = .03$; $z = 4.70$, $p < .001$; $CI$: .09; .24) were significant and independent mediators of this relationship. This model explains 49% of variance in authentic–durable happiness. Providing additional support for this model, the results did not provide a stronger support

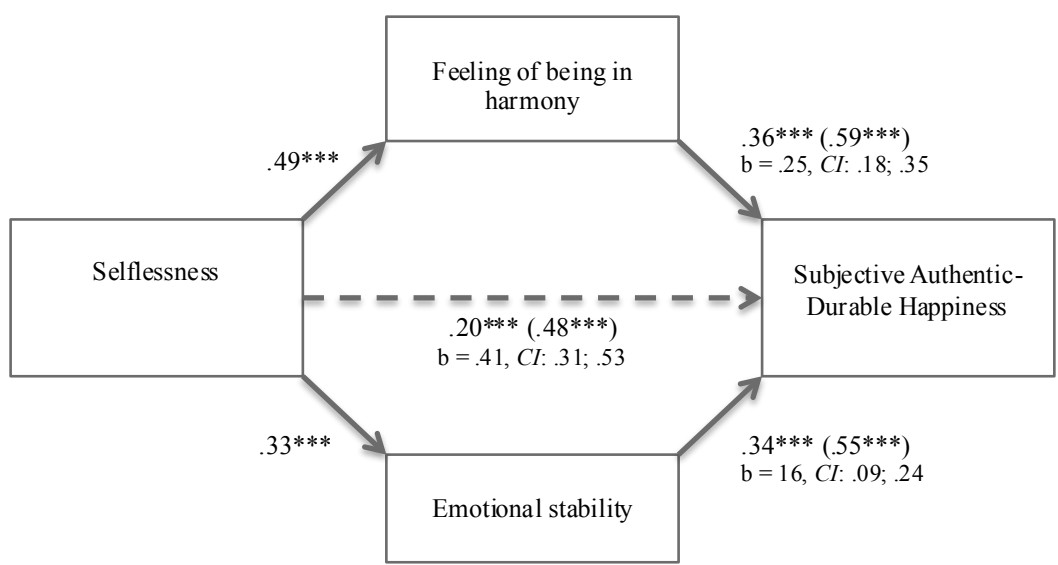

**Figure 2** Mediation model: feeling of harmony and emotional stability as two independent mediators of the relationship between selflessness and subjective authentic-durable happiness (study 2).

for the reversed model, in which harmony feeling and emotional stability mediated the relationship between authentic–durable happiness (IV) and selflessness (DV; $b = .15$, $se = .03$; $z = 4.51$, $p < .001$; $CI$: .08; .22). Moreover, in this reversed model, emotional stability did not mediate the relationship between the IV and the DV ($b = .02$, $se = .02$; $z = 1.01$, $p > .31$; $CI$: −.02; .07).

## Discussion

The results obtained in Study 1 are reproduced in this second study. First, the model in which selfless and self-centered psychological constructs load on two separate dimensions is supported by the data. Second, while selflessness was only related to subjective authentic–durable happiness, self-centeredness only correlated with subjective fluctuating happiness. Thus, using two samples, similar findings were found, suggesting that they are robust.

In addition, Study 2 reveals the main underlying psychological processes of the relationships between self-based psychological functioning and happiness. As predicted, the relationship between self-centeredness and subjective fluctuating happiness was fully mediated by afflictive affects. According to the SSHM, by favoring afflictive affects, self-centeredness would enhance fluctuations in term of happiness. Despite the impossibility to draw a causal conclusion due to the nature of the correlational design, the results of this second study are compatible with this interpretation. In addition, it was found that the relationship between selflessness and authentic–durable happiness was significantly mediated by both emotional stability and feelings of being in harmony. Interestingly, these two variables were two independent mediators. However, controlling for these two variables, the direct effect between selflessness and authentic–durable happiness remained statistically significant. This suggests that other processes are probably involved in this relationship. Future research may examine this possibility. Together, these results validate the hypotheses 3, 5 and 6.

While self-centeredness and selflessness are related to distinct happiness outcomes through distinct processes (i.e., afflictive affects and feeling of being in harmony), the role of emotional stability seems to be less exclusive. While emotional stability appears to be directly involved in the selflessness functioning, it also seems that it operates in self-centeredness, albeit at a more indirect level. While emotional stability did not significantly mediate the relationship between self-centeredness and fluctuating happiness, this construct was significantly related to both afflictive affects (around 12% of common variance) and fluctuating happiness (around 31% of common variance). This suggests a moderate overlap between self-centeredness and selflessness. While these two constructs did not correlate significantly in this study, it seems that both involved, at different levels, emotional (in)stability.

## GENERAL DISCUSSION

In recent decades, the study of well-being and happiness has gradually emerged as a central topic in psychology. One of the substantive questions underlying the research concerns the conditions for achieving happiness. The main current psychological theories emphasize the importance of several factors. For example, self-determination theory (*Deci & Ryan, 2000*) (see also *Ryan, Huta & Deci, 2008*) proposes that happiness and well-being are intimately linked to the achievement of three basic needs: autonomy, competence and relatedness. From the perspective of psychological well-being, *Ryff & Singer (2008)* highlight six key factors: self-acceptance, positive relationships, mastering one's environment, autonomy, having a purpose in life and personal growth. *Waterman (1993)* emphasizes the importance of carrying out activities that allow one to feel alive, to be engaged and fulfilled (e.g., *Waterman, Schwart & Conti, 2008*). Rather than focusing on the "psychological qualities" that achieve happiness, the SSHM takes into account the nature of the self and its resulting psychological processes. Specifically, it is argued that: (a) psychological functioning depends, at least in part, on the structuring of the self (more fundamentally of the "experienced self") and (b) the authentic–durable happiness, the state of plenitude and serenity, would be favored by selflessness, while fluctuations in term of happiness would be enhanced by self-centeredness. The results presented in this paper provide a preliminary support for these hypotheses. Let us see in more detail the main results of these two studies and their main limitations.

### Self-centeredness and selflessness are distinct psychological constructs

First, the theoretical model predicts the existence of two modes of psychological functioning (i.e., self-centeredness and selflessness) that tend to be opposed, but that they are characterized by qualitative distinct psychological processes. In two studies, results of factor analyses are compatible with this prediction. Each time, a two factors solution emerged with constructs related to self-centeredness loading on one factor and those assessing selflessness loading on a second factor. While these two factors were found to correlate negatively and modestly in Study 1 (i.e., around 6% of common variance), they were found to be statistically unrelated in Study 2. Thus, they are not strictly independent, but they are

[4] Across the two studies, the mean correlation between self-centeredness and fluctuating happiness was of .29 and the mean correlation between selflessness and authentic-durable happiness was of .42.

not simple opposites on a single continuum. While they tend to be opposed, they represent distinct psychological constructs. Consistently, the two types of happiness and the two self-based modes of psychological functioning indicate two relatively independent paths. While subjective fluctuating happiness was related to self-centeredness (path 1), subjective authentic–durable happiness was related to selflessness (path 2). The correlations were medium in size.[4] Finally, these two paths were explained by two distinct processes: afflictive affects (for path 1) and both feeling of being in harmony and emotional stability (for path 2). On the whole, these results are consistent with the two paths theoretical model and question approaches based on a single factor dimension, such as the quiet ego perspective (e.g., *Wayment et al., 2011*; *Wayment, Bauer & Sylaska, 2014*) or the hypo-ego conception (*Leary, Adams & Tate, 2006*; *Leary & Terry, 2012*), which are manifestations of self-centeredness. Since a low level of self-centeredness does not necessarily imply a high level of selflessness, it seems important not to limit the investigations to self-centeredness (i.e., low/high self-centeredness continuum). A more systematic differentiation between self-centeredness and selflessness processes seems to represent a positive avenue for future research. For example, both the quiet ego and the hypo-ego perspectives involve a sort of mindfulness attention: a ''detached awareness'' in the case of the quiet ego approach and a ''present-focused self-thought'' in the hypo-ego perspective. Based on the distinction between self-centeredness and selflessness, the theoretical model proposes that distinct attention processes may be involved in these two self-based modes of psychological functioning. While mindlessness through mental rumination (*Lo et al., 2014*), a wandering mind (*Killingsworth & Gilbert, 2010*) and the absence of meta-awareness (i.e., experiential fusion *Dahl, Lutz & Davidson, 2015*) would be involved in self-centeredness, selflessness would be more intertwined to a decentering attention (*Fresco et al., 2007*; *Hoge et al., 2015*; *Hadash et al., 2016*) and to mindfulness characteristics, such as attention to the present moment, acting with awareness, concentration, non-distraction and the non-judgment of experience (*Hadash et al., 2016*; *Baer et al., 2006*). Future research may examine this possibility.

## Self-based psychological functioning and happiness outcomes

Using the Schwartz's circumplex model of values (*Schwartz, 1992*), Dambrun and colleagues (*2012*) found that self-transcendence values (benevolence and universalism) were only related to authentic–durable happiness, while self-enhancement values (power and achievement) were only related to fluctuating happiness. The present results confirm these preliminary data with a more powerful approach in which selflessness and self-centeredness were assessed multi-dimensionally. In these studies, self-centeredness was inferred from egocentrism and materialism. Self-transcendence and other-connectedness were aggregated in order to infer selflessness. As expected, both dimensions were related to happiness outcomes in the expected directions. By fostering authentic–durable happiness, selflessness represents a promising way to enhance health and well-being. Using various scales assessing well-being and life satisfaction, authentic–durable happiness, particularly its inner-peace component, was the only robust predictor of a biological marker of stress (i.e., cortisol, see *Dambrun et al., 2012*). Inner-peace was associated with a lower level of cortisol, a steroid hormone involved in cardiovascular disease. Thus, insofar as selflessness increases
inner-peace, it could be a salutogenic factor for health. On the other hand, self-centeredness is likely implied in the development of health issues. In Study 2, both afflictive affects and subjective fluctuating happiness were significantly related to neuroticism (i.e., emotional instability), a personality trait strongly related to various psychological and health issues (e.g., *Charles et al., 2008*) and to mortality (e.g., *Mroczek, Spiro & Turiano, 2009*). Negative affects, closely related to afflictive affects, are associated with somatic symptoms and with an attention bias toward negative stimuli and threatening situations (*Denollet, 2013*; *Watson & Pennebaker, 1989*). Thus, it seems plausible that self-centeredness plays a role in health issues. Of course, much research is needed and research programs aiming at examining the relationships between self-based psychological functioning and various health issues must be profitably engaged.

## The underlying processes

An important contribution of this paper concerns the identification of some of the mechanisms which explain the relationship between self-based psychological functioning and happiness. Self-centeredness shares a non-negligible portion of variance with afflictive affects (i.e., 16%). As expected, these specific affects explained the relationship between self-centeredness and fluctuating happiness. It would be interesting, in future research, to identify the processes that explain the relationship between self-centeredness and these affects. Several variables could be involved, such as mindlessness (rumination, wandering mind), experiential avoidance (*Kashdan & Breen, 2007*) and susceptibility to ego-threat (*Leary & Terry, 2012*).

Selflessness was strongly and moderately related to two mediating variables: respectively, feeling of being in harmony (around 24% of common variance) and emotional stability (around 11% of common variance). In addition, these two mediating variables correlated moderately to authentic–durable happiness. These results can be added to the growing body of research confirming the traditional philosophies in which a state of harmony is a superior principle of the human existence that is intimately linked to happiness (e.g., *Kjell et al., 2015*; *Ip, 2014*; *Uchida & Kitayama, 2009*). As presented in the introduction, at least three levels of harmony can be distinguished: harmony with oneself, social harmony and harmony with the natural environment, including the universe. Future research may examine which of these components contributes the most to happiness (M Dambrun, 2016, unpublished data). It is also proposed to distinguish the cognitive and affective components of harmony. While the cognitive component (i.e., perceived harmony) is likely a prelude to affective harmony (i.e., feeling of being in harmony), it is posited that only the affective component would be primarily related to happiness. Because the cognitive component of harmony was not measured in this study, the question still remains open.

It is likely that benevolent affects (e.g., empathy, compassion, respect) are involved in the relation between selflessness and the two mediating variables. In a past study, it was found that the relationship between the self-transcendence inventory and subjective authentic–durable happiness was partially mediated by benevolent affects (*Dambrun & Ricard, 2012*). By improving interpersonal relations, benevolent affects may enhance social harmony. Moreover, pro-sociality is positively associated with emotional stability (e.g.,

*Carlo et al., 2012*). Thus, benevolent affects could explain, at least partially, why selflessness is related to both feelings of being in harmony and emotional stability.

## Limitations and future directions

Of course, these studies have some limitations. First, both self-centeredness and selflessness were respectively assessed through only two markers: egocentrism and materialism for the former and connectedness to other and self-transcendence for the latter. Thus, it would be important to replicate the current findings by adding additional markers. This would reinforce the present results. Second, these two studies are based on correlational devices and heterogeneous samples of French citizens. Such a sampling has the advantage of increasing the ecological validity of the results of these studies. However, because the present design is correlational, it is impossible to provide strong claims about causality. Future studies using experimental designs would increase the confidence in the causal direction between self-based psychological functioning (i.e., self-centeredness and selflessness) and happiness (i.e., fluctuating and authentic-durable). Encouraging results stem from a recent lab experiment, which reveals that selflessness elicits happiness via dissolution of perceived body boundaries *Dambrun (2016)*. Since the SSHM predicts reciprocal influences between self-based psychological functioning and happiness, experimental studies aiming at examining the influence of happiness on self-based psychological functioning would be also welcome. Nonetheless, these results provide significant empirical support for a part of the Self-centeredness/Selflessness Happiness Model (SSHM).

This theoretical model predicts that changes in the experience of the embodied self (from self-centeredness to selflessness) would lead to changes in emotional stability. The results that demonstrate the mediating role of emotional stability in the relationship between selflessness and authentic–durable happiness are compatible with this prediction, however, experimental studies are needed to specifically test this hypothesis, which challenges the view that personality traits are perfectly stable in adulthood (e.g., *McCrae & Costa Jr, 1994*). In this sense, the present results are consistent with the growing body of research revealing both changes in personality traits across the life course (e.g., *Roberts, Walton & Viechtbauer, 2006*; *Specht, Egloff & Schmukle, 2011*) and that personality traits and well-being can reciprocally influence each other over time (e.g., *Soto, 2015*).

Using the experience sampling method, it would be relevant to reproduce the present findings. According to the SSHM, self-centeredness and selflessness are not only related to distinct patterns of evaluated happiness, but also to distinct pattern of experienced happiness. Thus, it would be relevant to examine the hypotheses derived from this model using the experience sampling method. This approach minimizes the bias associated with the recovery of memories and those associated with the development of global judgments (e.g., judgments based on the most accessible memories, see *Kahneman, 1999*). In addition, these techniques provide continuous monitoring of longitudinal samples in the short or long term that can permit inferences regarding temporal relationships (e.g., *Steger, Kashdan & Oishi, 2008*).

### Funding

The author received no funding for this work.

### Competing Interests

The author declares that they have no competing interests.

### Author Contributions

- Michael Dambrun conceived and designed the experiments, performed the experiments, analyzed the data, contributed reagents/materials/analysis tools, wrote the paper, prepared figures and/or tables, reviewed drafts of the paper.

### Ethics

The following information was supplied relating to ethical approvals (i.e., approving body and any reference numbers):

Sud-EST VI statutory Ethics Committee (2014-CE36; IRB00008526).

### Data Availability

Dambrun, Michael (2016): Self-centeredness, selflessness and happiness: data2016. figshare.

https://doi.org/10.6084/m9.figshare.3467711.v3.

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
