# Peer review of "Self-centeredness and selflessness: happiness correlates and mediating psychological processes"

_PeerJ, doi:10.7717/peerj.3306_

## Round 0.1 · original submission · Major Revisions

I now have received two reviewers' comments. Although both reviewers expressed their interest in your study, several aspects of this manuscript should be revised to improve its clarity. Their observations are presented with clarity so I'll not risk confusing matters by belaboring or reiterating their comments. While I might quibble with the occasional point, I note that I regard the reviewers' opinions as substantive and well-informed.

I believe that all of the highlighted reservations require contemplation and appropriate attention in revising the document if it is to contribute appropriately to PeerJ and the extant literature. Please revise or refute according to the two reviewers' comments and provide a point by point reply in addition to the revised manuscript. In particular the concerns of Reviewer 1 (who recommended Rejection) must be addressed.

Tsung-Min Hung, Ph.D.
PeerJ editor
Distinguished professor
Department of Physical Education
National Taiwan Normal University

Reviewer 1 ·

Basic reporting

The manuscript has clear and professional English, and has good overall quality. However, the background and rationale was not convincingly proposed and some literature was not relevant, which makes research questions and hypotheses too unilateral and too ideal.
◆ In the Introduction section, though the authors explained SSHM very clear and definitely, the rationale was not robust. One of the reasons is that, the key studies repeatedly appeared in the manuscript were all from the same research groups. For example, in Page.4, the authors used only one REF(4), then directly proposed two self-functions. It’s not convincing enough. And the following description about these two self-functions and the possible mediators were all with the similar pattern.
◆ Besides, some REFs were either too irrelevant or too general. For example, in Page.6 Line. 139, it said that “Selflessness is closely related to self-transcendence (e.g.[18–20])”. Though REF 18-20 was indeed research about transcendence, we still did not know the similarity between these two concepts. And later in Line.146~, the authors proposed literature to explain the pro-social behaviors and positive outcomes (REF21-25); however, all these REFs are somehow too irrelevant to the focus of the present research.
◆ The same problem happened throughout the Introduction, and Discussion. For example, in Page.23 Line.565~, REF42-46 were proposed without proper explanation about the association with the research findings or research rationale. It, again, seems irrelevant and confusing.

Experimental design

In Methods, key information was not sufficient, such as the sampling method, validity of the measurements, & statistic analysis.
◆ How did these voluntary participants in Study 1 & 2 find out the information about this research? Besides their age, were any other background variables measured? Why did they need 24 hours to finish the survey? Is it possible to influence the results?
◆ The description about instruments was not sufficient. There should be validation processes to prove that Self-Transcendence + Connectedness = Selflessness, and Egocentrism + Materialism = Self-centeredness. Especially, as shown in Table.2, the correlation r between Self-Transcendence and Connectedness was only .31. Is it proper to say that Selflessness is the conjunction of Self-Transcendence & Connectedness? Is it proper to use an average score of these two concepts? (Same query could be applied to Egocentrism & Materialism, where r = .38)
◆ Another question is about the factor analysis. In Page.14 Line.338 & 340, the authors provided factor loading as .86, .78, .85, & .74. Only four numbers, so it means only average scores of four scales were utilized in the factor analysis, not all items of these scales? It's definitally not a proper way to exam the structure stability.
◆ Furthermore, the purpose of the FA was confusing. As mentioned above, it should have reasonable rationale about ST + C = Selflessness, and Ego + Mater = Self-centeredness.
◆ And is it necessary to repeat all the analysis of Study 1 allover again in Study 2? It's not clear why.
◆ The description of the validity of most measurements in this study was too brief. For example, Feeling of being in harmony was developed by Dambrun (REF40), but was this scale ever used in other studies?

Validity of the findings

The conclusions contained too many repeated descriptions about research results. It should be more discussion and comparison with other empirical studies and relevant literature.

Additional comments

◆ There's no information about statistic methods and software in both studies. Did the authors use SEM in Study 2? If so, where's the report of Goodness of Fit?
◆ There's no single REF in Discussion (Page.15-16 & Page.22-23).
◆ Page.20 Line.484: not sure what does „5000 samples“ mean.
◆ And is it necessary to repeat all the analysis of Study 1 all over again in Study 2? It's not clear why.

·

Basic reporting

1. The Introduction includes two distinct references to each hypothesis at different sub-sections. I suggest specifying each hypothesis once and only in the end of the Introduction. I think this will improve the readability of the article and will cut unnecessary repetitiveness in the Introduction.

2. In this article fluctuating happiness is only expected to correlate with self-centeredness, and authentic–durable happiness is only expected to correlate with selflessness. The author does not specify the reasons why fluctuating happiness is expected to display a non-significant correlation with selflessness and why authentic–durable happiness is expected to display a non-significant correlation with self-centeredness. Please clarify the reason why they are expected to be unrelated.

3. Some of the terms used in the Introduction need further clarification:
- The text in p. 4 lines 81-82 indicates that the self can be experienced at "different levels". Please clarify what are the different levels in which the self can be experienced
- In the text in p. 6 line 137 the author postulates that selflessness is a form of "self-based psychological functioning" but this doesn't seems logical. This seems to be an error since selflessness can’t be self-based. Please use a different expression (than self-based psychological functioning) or explain how “selflessness” is a form of self-based psychological functioning.
- In p. 6 lines 137-139 the author suggests that selflessness is based on a "small distinction" between self and environment as a whole, including others. This should be further explained as it isn’t clear from the text what a “small distinction" means. Does it refer to a sense of connection between self and environment?
- In p. 6 lines 140-142 the author suggests that selflessness would be primarily guided by the principle of "harmony", in the sense that the individual is facing an "optimal adjustment" with the different elements of the environment." Please clarify what the term “optimal adjustment” refers to. Is this a close relationship with the different elements of the environment?

4. In several occasions in the Introduction and Discussion some of the terms seem to be wrong or could be replaced by more appropriate terms. Here are some examples:
- P. 3 lines 67-68 say "It assumes that the self takes different structuration". I suggest to use the word structure.
- P. 3-4 lines 75-78 say “The experience of self is based on the processing of the sensory system (vision, kinesthetic, hearing, etc.), the treatment of the motor system (the underlying action) and the treatment of systems involving emotional, cognitive, and motivational operations.” I suggest to omit the words “treatment” from this sentence.
- p. 29 lines 674-675 says "these two studies are based on correlational devices". The word “devices” might be replaced by the word “methods”.

Minor comments:
- On p. 25 lines 609-612 the author may want to reference the following study in which selfless processing of experience was related to decentering, attention to the present moment and non-judgement of experience (Hadash, Plonsker, Vago, & Bernstein, 2016, Psychological Assessment).
- In Tables 3 & 5 please describe which variables are controlled under the "Partial β" column.
- In Figure 1b the lables of the processes "Self-Centeredness" and "Subjective Fluctuating Happiness" seem to be switched.
- On p. 18 line 440 the author refers to Table 3. The appropriate Table seems to be Table 4

Experimental design

Since the study was conducted on a French sample some of the measures probably needed translation from English to French. Please describe the procedure by which the measures were translated into French.
Where the measures back translated to ensure proper translation? (for a Translation and back-translation procedure see: Geisinger, 1994, psychological assessment)

Validity of the findings

1. The author used a principal component factor analysis with a varimax rotation. Varimax rotation is an orthogonal rotation which creates orthogonal/unrelated factors. However in the Introduction the author describes self-centeredness and selflessness as two distinct but related processes. Therefore, an oblique rotation seems to more appropriately fit the theory stated in the Introduction. Please explain the rational for using a varimax rotation. If the author believes that self-centeredness and selflessness are related processes he should consider reanalyzing the data using an oblique rotation which allow factors to be related.
2. On p. 24 lines 586-589 the author state that in the studies the two factors were unrelated or modestly related. This is an artifact of the varimax (orthogonal) rotation method that was used. This will always be the case when conducting a Varimax rotation. This should be clearly stated in the article.
3. p. 24 lines 592-593 says that subjective authentic–durable happiness was related only to selflessness. However the results from Table 3 suggest it was also related to self-centeredness in study 1. Therefore the word "only" in this sentence needs to be omitted.
4. On p. 26 lines 638-639 the author infers causality from correlational findings presented in the article ("the mechanisms by which self-based psychological functioning operates and affects happiness"). The wording needs to change as to reflect the correlational nature of these findings.

Additional comments

I appreciate the author's conceptualization and the operationalization of self-centered and selfless psychological fundtioning. I think this article is important for this field of research and deserves publication but requires several revisions before publication.

---

## Round 0.2 · Minor Revisions

I have now received two reviewers’ comment and both reviewers were generally satisfied with your reply and revisions from previous comments. However, both reviewers pointed out common issues in the introduction that require your further attention. There are other minor issues that also need some work. Please address these issues and provide a point by point reply in addition to the revised manuscript.

Tsung-Min Hung, Ph.D.
PeerJ editor
Distinguished professor
Department of Physical Education
National Taiwan Normal University

Reviewer 1 ·

Basic reporting

(Introduction section)
# L.58 "The objective of this study was to test three main assumptions of this model...". the sentence was put right after L.57, it's confusing that "this study" means "Dambrum & Ricard" or the present study. (especially combined with L.64).

# It's understandable that this paper was based on the seminal theoretical paper published in 2011, but from a reader's point of view, it seems that it contained too much information in Introduction section. For instance,"the Experienced Self" is truly important to understand the nature of SSHM, but not the main point of the present study. Maybe it could be explain by two or three concise summarized sentences, and the readers who have further interest on this topic could find more details through reference.

# Later in P.5~P.8, the information was overwhelmingly rich. Basically I think it contained the description of the relationship between Self-centeredness/Selflessness and happiness, and several different mediators respectively. In such case, could the subtitle be more specific? such as "The Self-Centered Psychological Functioning and the (possible) mediating effect of afflictive affects", "The Selfless Psychological Functioning and the (possible) mediating effect of....".

Besides, in P.5 it focused on the possible mediating effect of afflictive affects, but it seems that emotional stability and the feeling of being in harmony were all taken into account in the model (Fig.1). Is there any particular reason? (maybe I miss the information, if so, please ignore this question)

Experimental design

no comment

Validity of the findings

no comment

Additional comments

Overall, the comments were addressed detailedly and clearly. Though I still have some disagreement with the authors, it did not influence the excellent quality of this article. There are only few minor questions that might confuse the readers.

·

Basic reporting

The author was responsive and provided adequate answers to most of my queries. However, I believe that several paragraphs in the Introduction section still need to be revised to reduce its length, improve readability, and reduce inconsistencies and unnecessary repetitions. I still think this article is important for this field of research, and deserves publication but requires several revisions before publication:
1. The Introduction is lengthy and on some occasions the same ideas are presented repeatedly. For example: the idea that the nature of self-experience is the origin of the structure of the self is repeated in lines 68-69, 71-72, 82-83. Another example is the definition of fluctuating happiness as alternations between phases of pleasure and displeasure which is repeated in two consecutive sentences on p. 5 lines 115-119. As noted in my previous review, the revised version of the Introduction still includes two distinct references to each hypothesis. The author suggested these repetitions are helpful and pedagogical. I disagree with the author on this issue and still think they unnecessarily increase the complexity of the text. I suggest relating to each idea and hypothesis once to improve readability and cut unnecessary repetitiveness.
2. The second paragraph in the Introduction (lines 66-86) presents the idea that the nature of self-experience is the origin of the structure of the self. This paragraph is difficult to understand and its relations to the next sub-sections aren’t clear. Also, at the end of the paper the authors indicate that “the structure of the self and the “experienced self” were not measured” in this study! Since the study didn’t focus on the effects of self-experience on the structure of the self I suggest the author remove this paragraph. If the author chooses to retain this paragraph he should clarify how this subject is meaningfully related to the study, and which kinds of physio-psychological experience lead to distinct self-structures (permanent-separate/impermanent-connected). Examples would be very helpful here.
3. Following my suggestion the author changed the hypotheses concerning self-centeredness and fluctuating happiness (H2) and selflessness and durable happiness (H4). However this change was only integrated in the Introduction (page 10, lines 231-236). In other places in this manuscript the author still relates to the hypotheses from the previous version of this manuscript in which: (a) fluctuating happiness is expected to display a non-significant correlation with selflessness; and (b) authentic–durable happiness is expected to display a non-significant correlation with self-centeredness. I noticed this on pp. 10-11 lines 249-252 and on p. 15 lines 357-363. Please revise all references to this hypothesis throughout the manuscript consistently.
5. Following my suggestion the author changed the phrase “optimal adjustment” into “harmonious adjustment” in p. 6 line 142. However, the meaning of this term still isn’t clear enough. Please clarify what the term “harmonious adjustment” refers to. A definition and an example would be very helpful.
6. In the mediation analyses on pp. 21-22 the authors use the acronym “IC”. It seems that they refer to confidence intervals. If this is the case “IC” should be replaced with “CI” which is the correct acronym for confidence intervals.

Experimental design

The author refers to back translation procedure of several measures in the methods section using the text: “This scale was back translated”.
1. I believe the proper wording should be: “this scale was translated into French and then back translated”.
2. The author doesn’t state which translation procedure/guidelines were used. Proper translation is a very important and sensitive issue in this study since many of the scales used in this study were developed and validated in a different language! Please indicate which guidelines for translation/back translation were used (for examples: Geisinger, 1994, psychological assessment).

Validity of the findings

The author provided answers to all my queries regarding the validity of the findings, integrated my suggestions into the text and reanalyzed the data according to my suggestion.

---

## Round 0.3 · accepted · Accept

I have now received the reviewer's comment with a satisfactory feedback fro your revised manuscript. You and your coauthors have my congratulations. Thank you for choosing PeerJ as a venue for publishing your research work and I look forward to receiving more of your work in the future.

Tsung-Min Hung, Ph.D.
PeerJ editor

·

Basic reporting

The author provided adequate answers to all of my queries and integrated my suggestions into the text.

Experimental design

The author provided adequate answers to all of my queries and integrated my suggestions into the text.

Validity of the findings

The author provided adequate answers to all of my queries and integrated my suggestions into the text.

Additional comments

I believe this article provides an important contribution to field of self-centered and selfless psychological functioning, and deserves to be published in PeerJ.